# NF-ICP: Neural Field ICP for Robust 3D Human Registration

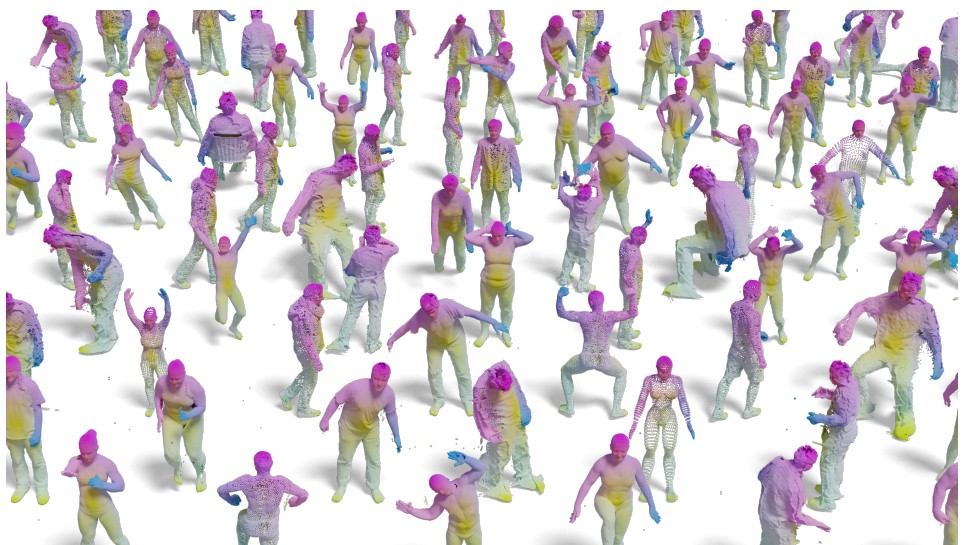

Figure 1: Our method provides reliable human registrations in disparate conditions. Our experiments involve more than 5K different shapes from more than 10 data sources and generalize to several situations: partial and cluttered point clouds, real scans, clothed humans, loose garments, identities out of distribution, and challenging poses. Training is performed only on raw synthetic SMPL shapes with no clothing or augmentation.

## Abstract

Aligning a template to 3D human point clouds is a long-standing problem crucial for tasks like animation, reconstruction, and most supervised learning pipelines. Recent data-driven methods leverage predicted surface correspondences; however, they are not robust to varied poses or distributions. In contrast, industrial solutions often rely on expensive manual annotations or multi-view capture systems. In this work, we present NF-ICP, a method that, for the first time, generalizes well on a large set of challenges, including complex poses, clothed humans, and noisy scans. Leveraging the large MoCap dataset AMASS, we learn a neural field model to predict the direction towards the localized SMPL vertices on the target surface. Such neural field leads to a reasonable initialization, but the resulting template often does not overlap with the target surface. NF-ICP exploits a classical Iterative Closest Point objective adapted to our model to quickly fine-tune the model, resulting in a significantly improved template to target surface overlap. NF-ICP constitutes a simple and computationally efficient registration method that significantly improves over public benchmarks and solidly surpasses the state of the art. We will release code and checkpoints in `link`.

## 1 Introduction

Registration of 3D surfaces is a significant area of research in Computer Vision, playing a crucial role in analyzing shape collections, pattern discovery, and statistical model training. Among all the 3D surfaces, human models are of particular importance. It is worth noting that human registration has enabled the development of today's standard human parametric model (SMPL from Loper et al.

(2015)) and also facilitates several downstream tasks such as animation, virtual try-on, and is the driving force behind virtual and mixed reality.

However, human registration presents several challenges. Human articulations can take various configurations, leading to blind spots and geometry gluing (for instance, due to self-contact), even when hundreds of cameras are involved. Fine-grained details characterize human identities, and their reconstruction is essential to represent human diversity. Finally, the acquisition process is noisy, especially when low-resource settings (like Kinect depth scans) or even customer-level devices (such as applying NeRF-based systems like Luma AI to smartphone RGB videos) are used.

Scholars approach shape registration intrinsically or extrinsically. Intrinsic approaches, favored by Computer Graphics (Ovsjanikov et al. (2012); Sun et al. (2009); Melzi et al. (2019b); Marin et al. (2020a)), obtain pair-wise correspondences between shapes, restricting solutions to the object's surfaces and guaranteeing invariance to rigid deformations. These methods are often impractical for real data, as clutter and noise often ruin their theoretical premises. Extrinsic registration pipelines, popular in Computer Vision research (Bogo et al. (2014); Zuffi et al. (2017a); Li et al. (2017a;b); Romero et al. (2022)), rely on templates as regularizers. However, these pipelines require expert annotators, multi-camera views, and priors for specific settings, which makes them inapplicable in many use cases. The final template deformation may not lie on the target surface even with robust learned priors. A common practice to refine template alignment is to iterate between Euclidean correspondence and registration until convergence, as done in the popular Iterative Closest Point (ICP) of Besl & McKay (1992b) and its variants like the one from Li et al. (2008). However, the pairing step is often sensitive to initialization and geometrical noise, leading to local minima.

Our work presents an iterative method combining intuitions from different approaches to produce a simple yet effective solution. Inspired by the Learned Vertex Descent (LVD) method of Corona et al. (2022), we train a Neural Field (NF) that, given a target shape, predicts the offsets from a query point in $\mathbb{R}^3$ toward template registration vertices. The output offsets are ordered as the template vertices, and their application aligns the template to the target. However, LVD produces reliable results only on samples within the training distribution. Therefore, we refine the NF deformation using a self-supervised task that, *at inference time*, improves network generalization to unseen data. We alternate two steps: first, we use the NF to produce a correspondence between the template and the target. Second, we optimize the NF offset prediction to adhere to this correspondence. Concretely, we query the NF directly on *the vertices of the target shape*. For each point on the target, we pair it with the template vertex *corresponding to the predicted offset with the minimum norm*, i.e., the closest template point indicated by the NF. Secondly, we sum the retrieved smallest offsets for all the target points, and we use this as a loss to *fine-tune the NF by backpropagation*. This simple procedure is based on ICP principles: NF deformation suggests a correspondence, which we use to update the deformation. Our pairing step exploits data-driven prior learned by the NF, making it robust to geometrical noise. Our approach is the first refinement specifically designed for NF, takes a few seconds, and improves the final registration from 18% to 30%.

We apply our approach to a novel localized variant of LVD trained on the large MoCap dataset from Mahmood et al. (2019) (AMASS). Using a *single network*, we provide a ready-to-go method tested on more than 5k shapes and 10 different data sources; Figure 1 shows a subset of these, and colors encode the correspondence provided by our registration. We stress it with an unprecedented variety of poses, identities, and real-world challenges far from training distribution (e.g., garments, noise point clouds, clutter, partiality). We largely improve state-of-the-art benchmarks for human body registration for methods trained on similar assumptions and surpassing approaches with more robust priors. We will release our code and weights of the network, providing a tool that, out of the box, offers robust human registration in *less than a minute*. In summary, our contributions are:

1. *NF-ICP*: A novel self-supervised fine-tuning procedure to improve the registration predicted by a neural field; it is the first iterative refinement approach for a neural field, which largely improves its registration performance

2. *Human registration*: we deploy a complete registration pipeline for human bodies with a novel localized variant of LVD, reaching a new state of the art on different benchmarks and providing an unprecedented robust approach

3. *Code and data release*: all the code for processing, training, and evaluation, together with network weights and data will be released as a useful tool for researchers.

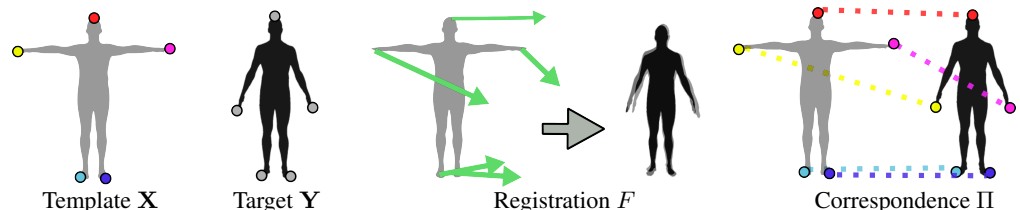

Figure 2: 2D visualization of components used in our approach. We use 2D examples to make our explanation more didactical, while we remark that our method works with 3D point clouds.

## 2 RELATED WORK

Shape correspondence has a vast literature, and while we list the works that mainly inspired us, we point to Deng et al. (2022) for a recent and extensive survey.

**Shape matching.** A straightforward idea to solve the correspondence problem is to compute a space for the vertices of the shapes where they are naturally aligned (canonical embedding). Classical approaches in this category are the multi-dimensional scaling from Bronstein et al. (2006), and descriptors proposed by Sun et al. (2009) (HKS), Aubry et al. (2011) (WKS), and Salti et al. (2014) (SHOT). Their dependence on input geometry does not generalize well in the presence of noise. Recently, Marin et al. (2020b) proposes a deep learning variation as a baseline. Still, such a unified representation is challenging to learn for non-rigid shapes. Works like Cao et al. (2019); Cao et al. (2017); Kim et al. (2021) propose canonicalizing human shapes by inferring a common skeleton from external views, but they suffer from occlusions. A recent trend (Marin et al. (2020b); Huang et al. (2022); Jiang et al. (2023)) is to learn a representation where a linear transformation can align shapes. These methods rely on pseudo-inverse computation at training time, which hardly scales and produces numerical instability. Their inspiration can be found in the Functional Maps framework of Ovsjanikov et al. (2012), an elegant and theoretically grounded formulation which opened to a set of regularization and variations (Nogneng & Ovsjanikov (2017); Ren et al. (2018); Marin et al. (2023)), as well as refinements (Huang et al. (2020); Melzi et al. (2019b)). Mainly based on the eigenfunctions of the Laplace-Beltrami Operator, its applications are limited to clean meshes and unrealistic partialities (Melzi et al. (2020); Rodolà et al. (2017); Cosmo et al. (2016)).

**Shape Registrations.** A widely explored approach to recover the correspondence is solving the alignment problem by registering the pair of shapes in $\mathcal{R}^3$. A fundamental algorithm for the rigid case is ICP from Besl & McKay (1992b), which aligns two shapes by iterating between the deformation and the correspondence. The simplicity of ICP made it broadly used, but its sensitivity to noise and initialization may limit the convergence toward a local minimum. Countless variations have been proposed; for example, Bouaziz et al. (2013) and Chetverikov et al. (2002) address the sparsity and noise, respectively, but they still require careful design choices. Interestingly, Wang & Solomon (2019) and Yu et al. (2023) propose a deep learning approach to provide a more reliable correspondence. Still, they are both limited to rigid deformation. Instead, iterative algorithms to recover non-rigid deformation like Li et al. (2008) often rely on optimizations that trade some rigidity for a more expressive deformation. The Coherent Point Drift proposed in Myronenko & Song (2010) exploits a probabilistic formulation, which got a new life following probabilistic shapes representations (Hirose (2020; 2022)). Deep learning follow-ups from Li et al. (2022) and Wang et al. (2019) tried to learn the deformation from data. However, their learning of point-wise features assumes limited non-rigid deformations and the absence of clutter. Templates provide strong regularization for real scenarios (Zuffi et al. (2017b); Hesse et al. (2018)). This promising direction has been widely investigated (Groueix et al. (2018); Bhatnagar et al. (2020); Trappolini et al. (2021); Sundararaman et al. (2022)) — but the final registration rarely aligns on the target surface.

**3D Human registration.** The classic work of Bogo et al. (2014) uses expert feedback and controlled setups to lead 3D registration. More flexible, the stitched puppet of Zuffi & Black (2015) automatically solves piece-wise optimization of the local body parts, but gluing them leads to major artifacts. Marin et al. (2020a) introduced FARM, an automatic method with 3D landmark detection, which Marin et al. (2019) extended for high resolution. Both rely on the Functional Map of Ovsjanikov et al. (2012) and struggle with non-watertight meshes. With the advent of learning, the seminal work of Groueix et al. (2018) proposed a simple yet effective autoencoder architecture. Its global nature motivated follow-up works of Deprelle et al. (2019) and Trappolini et al. (2021) to learn local and attention-based relations, but both suffer from clutter. Kim et al. (2021) pro-

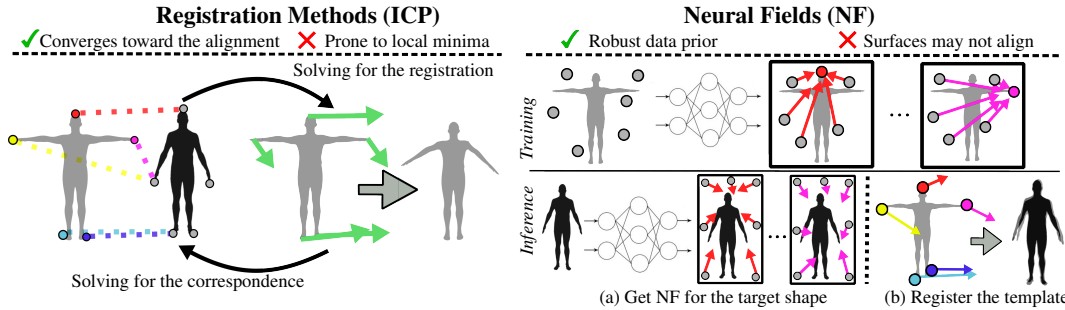

Figure 4: Schema of ICP and Neural Fields. Iterative approaches refine their prediction and converge to a minimum, which is often local due to the sensitivity of the pairing step. NF uses robust data prior and rich representations, but surfaces may not align since it is bound to training distribution.

pose Deep virtual markers (DVM), using synthetic multi-view depth representation, that requires a demanding manually annotated training set, and suffer from occlusions and self-contact. Instead of directly predicting the position of the points, Sundararaman et al. (2022) output offsets.

While similar in principle to predicting space positions, this output representation is richer and predisposed to regularization. Such regularizations are expensive, limiting applicability to small training distributions. Corona et al. (2022) recently proposed Learned Vertex Descent (LVD), which instead regularizes the training using an even richer representation for the template deformation. Their idea is to train a network that, given an input shape, produces an NF defined all over $\mathbb{R}^3$. Querying the NF to a 3D point predicts the ordered offsets toward all the registered template vertices. Supervising the training on every 3D location toward all the template vertices produces a robust data-driven prior that benefits from large-scale datasets. However, LVD is

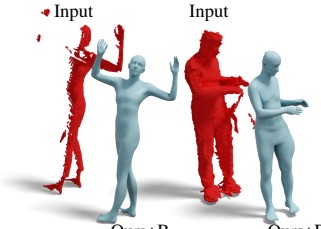

Figure 3: Input and results of our method on some challenging shapes from Bhatnagar et al. (2022) and Ma et al. (2020).

also highly dependent on the training distribution. Finally, registration methods often rely on a refinement based on Chamfer distance. While effective, similarly to ICP, this requires an excellent initialization to fit details correctly, especially in the presence of noise or clutter in the target point cloud, and does not exploit the data-driven prior.

## 3 BACKGROUND

### 3.1 POINT CLOUD REGISTRATION

3D Point cloud registration is the process of spatially aligning a template with an unsorted target point cloud while respecting their semantics (see Figure 2 for a 2D intuition of the elements described in this paragraph). Given a template with ordered points $\mathbf{X} \in \mathbb{R}^{m \times 3}$ and an unsorted target point cloud $\mathbf{Y} \in \mathbb{R}^{n \times 3}$, the goal is to recover a deformation $F : \mathbf{X} \to \hat{\mathbf{X}} \in \mathbb{R}^{m \times 3}$ such that $\hat{\mathbf{X}}$ aligns with $\Pi(\mathbf{Y})$ under a permutation $\Pi : \mathbf{Y} \to \hat{\mathbf{Y}} \in \mathbf{Y}^m$, which encodes the semantic correspondence:

$$\|F(\mathbf{X}) - \Pi(\mathbf{Y})\|_F = 0. \tag{1}$$

However, the real case is more complex as the correspondence $\Pi$ is not always bijective due to different numbers of points, partial views, noise, and clutter. As a result, often Equation 1 does not have a ground-truth solution and is solved by optimization. Figure 3 shows examples of partial and cluttered input (red point clouds), as well as the output of our method (grey meshes).

### 3.2 ITERATIVE CLOSEST POINT

Among the possible optimizations to solve Equation 1, the Iterative Closest Point (ICP) algorithm proposed by Besl & McKay (1992a) is one of the most popular approaches. The main idea is to solve the correspondence and the deformation iteratively such that they refine each other till convergence. Starting from an initial configuration of the two shapes (*i.e.*, $\widetilde{F}(\mathbf{X}) = \mathbf{X}$), ICP recovers the correspondence using an Euclidean nearest-neighbor pairing:

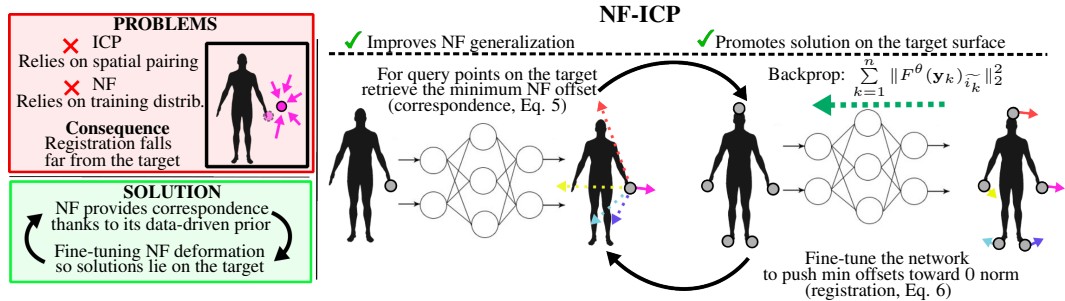

Figure 5: The intuition behind NFICP. We identify the limitations of the previous methods and we propose a new iterative fine-tuning for NF. We alternate between two steps: we query points on the target shape and we identify a correspondence with the template vertices (Equation 5). Then, we refine the NF by minimizing the predicted offsets and promoting template vertices to converge toward the corresponding points (Equation 6).

$$\widetilde{\Pi} = \arg\min_{\Pi} \|\widetilde{F}(\mathbf{X}) - \Pi(\mathbf{Y})\|_2^2. \tag{2}$$

Next, ICP updates $\widetilde{F}$ by minimizing the point-to-point distance with a least-squares objective:

$$\widetilde{F} = \arg\min_{F} \sum_{i=1}^{m} \|F(\mathbf{x}_i) - \widetilde{\Pi}(\mathbf{Y})_i\|_2^2. \tag{3}$$

These two steps are iterated multiple times until convergence. We report a graphical intuition of ICP on the left of Figure 4.

**ICP Limitations.** The primary limits of iterative registration methods based on the geometry are the sensitivity of the correspondence step in the presence of noise. The Euclidean nearest-neighbor pairing is undesirable both for computational and stability reasons and requires ad-hoc mitigation strategies in the presence of outliers (Gelfand et al. (2003); Men et al. (2011); Yew & Lee (2020)). ICP converges at a global minimum only when the two shapes are already close enough; otherwise, it gets trapped in local minima with terrible misalignments.

### 3.3 NEURAL FIELDS FOR REGISTRATION

Neural Fields (**NF**) have been proven to be a powerful representation of 3D geometry. The recent survey proposed by Xie et al. (2022) defines an NF as a quantity defined all over a domain coordinates (a field), parametrized by a neural network. Corona et al. (2022) firstly propose their use for solving the registration problem, calling this procedure Learned Vertex Descent (LVD). The idea is to train a neural network $F^\theta : \mathbb{R}^3 \to \mathbb{R}^{n\times 3}$ that, for any point in $\mathbb{R}^3$, outputs the ordered offsets toward the $n$ vertices of the desired deformed template $\hat{\mathbf{X}}$ (Figure 4, right top). At inference time, the network produces the NF for a target shape (Figure 4, bottom right (a)). The location $\hat{\mathbf{x}}_i$ of the $i$-th registered template vertex is obtained by querying the NF on $\mathbf{x}_i$ and following the $i$-th predicted offset $F^\theta(\mathbf{x}_i)_i$ (Figure 4, bottom right (b)):

$$\hat{\mathbf{x}}_i = \mathbf{x}_i + F^\theta(\mathbf{x}_i)_i \tag{4}$$

**NF Limitations.** As is often the case, data-driven approaches like LVD are bound to their training distribution. Considering the variety of human identities and poses, further complicated by external clutter (*e.g.*, garments, noise), it is not rare that network predictions miss the target surface.

## 4 METHOD

**Overview.** Our registration method takes a 3D point cloud. The input is passed to a backbone network that obtains NF as described in Section 3.3. Before evaluating the NF, we refine it using NF-ICP, a self-supervised task that iteratively improves the backbone. Then, we use the updated NF to register the template to the input point cloud. The registration output by the network is further refined by Chamfer registration and, if high-frequency details are required, also by local vertex

displacement. In the following, we introduce the NF-ICP procedure and present the detailed steps of our registration pipeline.

## 4.1 Neural Field ICP

**Intuition.** NF defines a deformation field that can be queried all over $\mathbb{R}^3$, also on the target shape. Template registration should align with the target, so we expect that queries of the NF on the target vertices produce one or more offsets with norms close to 0. We propose an iterative self-supervised task to promote this desirable property.

**NF-ICP** Given a target shape, we iterate two steps. First, we *solve for the correspondence*. We sample points $\mathbf{y_k}$ over the target shape and use them to query the NF. We pair every query point with the template vertex that has the predicted offset with the minimum norm in that point:

$$\widetilde{i}_k = \arg\min_i \|F^{\widetilde{\theta}}(\mathbf{y}_k)_i\|_2^2, \tag{5}$$

which replaces the Euclidean $\arg\min$ of Equation 2 with one suggested by the network data prior.

Secondly, we *solve for the registration*. We sum the minimum offsets $F^\theta(\mathbf{y}_k)_{\widetilde{i}_k}$ for all the query points $\mathbf{y}_k$, and we backpropagate this amount to update the NF parameters:

$$\widetilde{\theta} = \arg\min_\theta \sum_{k=1}^n \|F^\theta(\mathbf{y}_k)_{\widetilde{i}_k}\|_2^2. \tag{6}$$

Namely, we refine the predicted field such that the deformation converges toward the estimated correspondence (i.e., it predicts a 0 offset for the associated template vertex). Iterating between these two steps refines the backbone network. We call this procedure *Neural Field ICP* (**NF-ICP**); we report a didactic intuition in Figure 5. We stress that NF-ICP is a self-supervised fine-tuning performed *at inference time*, that does not require further data or supervision.

## 4.2 NF-ICP in action: Human Registration

**Input.** Given an input point cloud, we only assume a rough estimation of the Y-axis, while the points can be rotated in any other direction. Such assumption is standard in human body registration pipelines like the ones from Groueix et al. (2018) and Trappolini et al. (2021), and can be easily approximated, for example, by computing robust landmarks for the shapes as in Marin et al. (2020a).

**Neural fields: multi-head LVD.** We explore different types of NF. We begin by training *OneShot*, a baseline that predicts offsets toward the registered template vertices for every point in $\mathbb{R}^3$ in a single pass. Next, we examine $\mathbf{LVD}_1$, which uses the original formulation of Corona et al. (2022) and predicts smaller offsets. The template vertices are moved until convergence. Finally, we propose a novel variant of LVD that employs multiple MLP output heads, each specialized in predicting offsets only for vertices in a local region of the template. We obtain the regions by using spectral segmentation on the template shape Liu & Zhang (2004), which provides a sound geometrical structure. We discovered that 16 local segments yield the best results, and we call our variant $\mathbf{LVD}_{16}$. In the case of experiments on MANO, we directly use the pre-trained network weights released by Corona et al. (2022), proving that NF-ICP can be applied on top of different architectures out of the box. We refer to supplementary for technical details on the backbones, segments, and validation of design choices.

**NF-ICP and template fitting.** Given a target point cloud, we use the network to obtain the NF, and we fine-tune it by iterating between Equation 5 and Equation 6. We update the network weights 20 times with a learning rate of $1e - 5$. The whole procedure takes around 3 seconds. Then, we use the NF to move the points of the template, and we obtain the 690 points of the registered template. Finally, we fit a complete SMPL model to the 690 points, using a pose prior penalization proposed in Pavlakos et al. (2019). We name the results obtained with $\mathbf{LVD}_{16}$ + NF-ICP as **Ours**.

**Refinement.** A common practice is to refine the template by optimizing its parameters with Chamfer distance. When we assume bijectivity between the template and target, we use the bidirectional one; otherwise, as done in Bhatnagar et al. (2020), we compute Chamfer distance in a single direction. Finally, in cases where a high resolution is required, following the spirit of Marin et al. (2019), we use SMPL+D to catch the finer details of the target, regularizing the displacements with a Laplacian energy Gao et al. (2020). When one or both of these refinements are applied, we will refer to them as **+R** after the method name. We report further details in the supplementary

| Method | MANO Error | | Reconstruction to Scan | | Scan to Reconstruction | |  |
| | Joint [cm] | Vertex [cm] | V2V [mm] | V2S [mm] | V2V [mm] | V2S [mm] | |
| --- | --- | --- | --- | --- | --- | --- | --- |
| $LVD_1$ | 9.6 | 12.3 | 5.73 | 5.73 | 8.16 | 6.43 | |
| $LVD_1$+NF-ICP | **7.9** | **10.7** | **4.73** | **4.70** | **6.37** | **4.15** | |

Table 1: Hands registration. We apply NF-ICP directly to the pre-trained network released by Corona et al. (2022). NF-ICP works out of the box and improves all the metrics. On the right is a qualitative comparison.

## 5 RESULTS

**Remark.** All shown results on humans are obtained starting from the same network weights. We are unaware of any other method that provides an extensive evaluation and generalization, such as the one shown here and in the supplementary. The experiments stress our method quantitatively and qualitatively on the FAUST (Bogo et al. (2014)) and SHREC19 (Melzi et al. (2019a)) challenges, which include shapes from 10 different datasets. We also show results on partial point clouds from CAPE (Ma et al. (2020)) and noise one from BEHAVE (Bhatnagar et al. (2022)), 3D human reconstructions obtained with (Luma AI), cases significantly out of distribution from the Scan The World Project Project and TOSCA (Bronstein et al. (2008)). In the teaser, we reported results for challenging poses from SCAPE (Anguelov et al. (2005)), clothed humans from Twindom (Twindom Dataset), and RenderPeople (renderpeople). In supplementary we report further results on Dynamic-Faust (Bogo et al. (2017)) and the recent HuMMan dataset (Cai et al. (2022)).

**Templates.** We consider the human SMPL model sampled at 690 vertices as a template; hence, our NF will provide $690 \times 3$ values for each point. This subsample speeds training and inference up while we have enough information to fit a complete SMPL model later. For the hand-fitting experiment, we use the MANO model proposed in Romero et al. (2017).

**Data.** To train the NF, we leverage the large MoCap AMASS dataset of Mahmood et al. (2019), adopting the official splits. The train set comprises roughly 120k SMPL+H Romero et al. (2017) shapes animated with motion-captured sequences. We train for 10 epochs ($\sim$ 157k steps). We report implementation details in supplementary. For ablations, we use the training set of the FAUST challenge. We will consider both cases in which the input shape is the registration (**Faust**$_R$) or the real scan (**Faust**$_S$).

**Baselines.** We compare against methods using the same feature extractor and training set. First, as described in the previous section, we consider the original LVD formulation (**LVD**$_1$). We also reimplemented the Universal embedding baseline proposed by Marin et al. (2020b). This approach learns a high-dimensional embedding where shapes are naturally aligned, and the correspondence is obtained by computing the nearest neighbor in that space. We consider a 60 dimensional embedding (**Uni-60**) as proposed by Marin et al. (2020b), and also a 2070 one (**Uni-2070**) to have similar output dimensions of our method (690 offset of 3 dimensions).

**Metrics.** For the test on the FAUST training, FAUST challenge, and BEHAVE the error is measured in centimeters as the Euclidean distance from the predicted point position and the ground truth. On SHREC19, the error is the normalized geodesic distance from the ground truth Kim et al. (2011). The Metric of MANO experiments follows the same protocol proposed by Corona et al. (2022), as vertex-to-vertex (V2V) and vertex-to-surface (V2S) Euclidean errors.

**Computational timing.** We conduct our experiments on a computer equipped with a 12-core CPU AMD Ryzen 9 5900X, an NVIDIA GeForce RTX 3080 Ti GPU, and RAM 64GB. The NF-ICP refinement requires $\tilde{3}$–4 seconds, and the entire registration for a single shape no more than 60 seconds. The variance is mainly caused by the initial parsing of the input point cloud.

### 5.1 ABLATION STUDY

In this section, we apply NF-ICP to refine different backbone NF, also on a different domain (hands), and compare against baselines trained on the same data. Further ablations on registration components are reported in the supplementary material.

**Different backbones.** We are interested in validating different possible NF backbones and the effect of NF-ICP on them. Table 2 shows the results on the three considered backbones. Our proposed variant $LVD_{16}$ performs the best, showing that the proposed localized MLPs are effective. Secondly, NF-ICP consistently *improves all the NF*, proving its generality.

**Different Domain.** To further prove the generality of NF-ICP, we test its applicability to a different domain. We replicated the experiment on the hands' registration of Corona et al. (2022), using publicly released pre-trained weights and evaluating on the same protocol. We report the errors in Table 1. NF-ICP improves both vertices and joint estimations.

**Same training setting.** Table 3 reports the comparison with methods trained on the same dataset and the same feature extractor. We notice that the Universal baseline is a challenging competitor, able to surpass recent methods (Corona et al. (2022)), but NF-ICP provides a key advantage. We remark that Uni-2070 computes nearest neighbors in a highly dimensional space, which is particularly expensive.

|  | $\text{FAUST}_R$ | $\text{FAUST}_S$ |
|---|---|---|
| OneShot | 4.45 | 3.34 |
| OneShot + NF-ICP | *3.64* | *3.15* |
| $\text{LVD}_1$ | 4.78 | 3.54 |
| $\text{LVD}_1$ + NF-ICP | *3.40* | *2.81* |
| $\text{LVD}_{16}$ | 4.35 | 3.11 |
| $\text{LVD}_{16}$ + NF-ICP (Ours) | ***2.97*** | ***2.55*** |

Table 2: NF-ICP enhance different NF; Our variant $\text{LVD}_{16}$ performs the best, and NF-ICP improves all the NF.

|  | $\text{FAUST}_R$ | $\text{FAUST}_S$ |
|---|---|---|
| Uni-60 (Marin et al. (2020b)) | 3.52 | 2.66 |
| Uni-2070 (Marin et al. (2020b)) | 3.49 | 2.62 |
| $\text{LVD}_1$ (Corona et al. (2022)) | 4.78 | 3.54 |
| $\text{LVD}_{16}$+NF-ICP (Ours) | 2.97 | 2.55 |
| Ours + R | **1.76** | **1.85** |

Table 3: Comparison with the baselines using the same training data and feature extractors. NF-ICP provides a crucial improvement.

## 5.2 PUBLIC BENCHMARKS

**FAUST Challenge.** We test our pipeline on real data from the two FAUST challenges[1] from Bogo et al. (2014). The scans are of 10 subjects in 20 poses for 200 test samples, with missing regions and clutter due to the noise in the acquisition process. Table 4 reports the error results from the leaderboard in millimeters. We list the methods with similar assumptions to ours in the upper part of the table. We achieve an improvement of around 20% and 14% on the INTRA-subject and INTER-subject challenges, respectively. We also outperform DVM Kim et al. (2021), which requires a manually artist-annotated dataset for training, the render of 72 synthetic views of the target for inference, and *have seen FAUST shapes at training time*, while we do not. We also report the evaluation of our method disabling the NF-ICP refinement. We appreciate the fundamental role of NF-ICP in aligning details to obtain high-quality correspondence. On the right of Table 4, we also visualize qualitative results, reporting input, output, and point-to-point error.

**SHREC19.** To stress our generalization, we test our method on the more recent and challenging dataset SHREC19 Melzi et al. (2019a). This database is a collection of 44 shapes from different datasets, facing various challenges: holes, out-of-distribution identities, disconnected components (*e.g.* earrings), clutter, and different densities (shapes have from 5k to 200k vertices). The ground truth on this data is provided by a robust registration method that exploits surface information to regularize the results. We report the results in Table 5 and the leaderboard from Sundararaman et al. (2022), showing our method significantly outperforms previous state-of-the-art and cuts errors by more than half. On the right of Table 5, we visualize different examples and registrations from our method, using a high-frequency function transfer to emphasize the semantic consistency of our registration. Our method fits non-isometric changes and challenging poses well.

## 5.3 APPLICATIONS

**Real point clouds.** Addressing the challenges posed by partial and cluttered point clouds is an important area of research with real-world applications. It is also a fascinating challenge to test the robustness of NF-ICP: our self-supervised task has no clue to distinguish points belonging to the human from those that are outliers. We test our method on the BEHAVE dataset proposed in Bhatnagar et al. (2022), which contains point clouds of humans interacting with objects. These point clouds are obtained by merging Kinect multi-view depths and present significant noise. Although it includes masks to isolate the human, portions of the objects and background are often captured, the views are fused incorrectly, and human limbs may be missing due to occlusions. We uniformly sample 3320 frames from the sequences, register the fused depth point cloud, and compare the error with the provided ground truth registrations. These latter were obtained by Bhatnagar et al. (2022) using a complex pipeline that requires RGB information and manual intervention. This ground truth provides a good upper bound to our method, which *is automatic and only relies on the 3D geometry of the input*. Results are displayed in Figure 3 and Table 6. Contrary to previous methods, which can address clutter and partiality only if seen during training time (Attaiki et al. (2021), Huang et al.

---

[1]https://faust-leaderboard.is.tuebingen.mpg.de/leaderboard

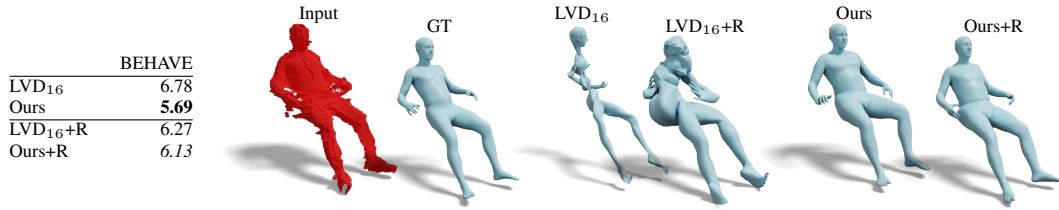

| | INTRA | INTER |
|---|---|---|
| Loopreg (Bhatnagar et al. (2020)) | 1.35 | 2.66 |
| Elementary (Deprelle et al. (2019)) | 1.74 | 2.58 |
| UD2E-Net (Chen et al. (2021)) | 1.51 | 3.09 |
| Ours+R | **1.06** | **2.26** |
| LVD$_{16}$+R | 1.11 | 2.48 |
| DVM (multi-view)† (Kim et al. (2021)) | 1.19 | 2.37 |

†: Has seen FAUST training set

Table 4: Results on INTRA- and INTER-subject FAUST challenges. On the left, we report the results from the public leaderboard. On the right, for each shape, we report the target surface, our output, and point-to-point error visualized as a heatmap on the target surface. Noticeably, no distance exceeds 1cm.

| | SHREC19 |
|---|---|
| LIE (Marin et al. (2020b)) | 15.1 |
| GeoFMAP (Donati et al. (2020)) | 11.2 |
| CorrNet3D (Zeng et al. (2021)) | 9.6 |
| 3DCoded (Groueix et al. (2018)) | 10.3 |
| Transmatch (Trappolini et al. (2021)) | 6.1 |
| ReduceBasis (Sundararaman et al. (2022)) | 4.8 |
| Ours + R | **2.27** |

Table 5: Result on challenge proposed by Melzi et al. (2019a). For each shape, we report the target surface and our output. We also visualize a transfer of a high-frequency function, showing the semantic correspondence.

| | BEHAVE |
|---|---|
| LVD$_{16}$ | 6.78 |
| Ours | **5.69** |
| LVD$_{16}$+R | 6.27 |
| Ours+R | *6.13* |

Table 6: Results on BEHAVE dataset. On the left, quantitative registration error on 3320 frames; NFICP improves on average of 17,6% over the NF results in 80% of the frames. On the right is a qualitative result; the Chamfer-based refinement can even worsen the results, but our method brings it closer to a better minimum.

(2021)), our NF-ICP approach enables compelling generalization without this requirement. NF-ICP enhances the backbone NF by 17.6%. NF-ICP improves the results in approximately 80% of the frames. Interestingly, we observe that the degraded geometry of the input negatively impacts Chamfer refinement, whereas the data-driven approach of NF-ICP is more robust and can even initialize Chamfer closer to a better minimum. The supplementary material contains further visualization and discussion on failure modes.

**Human Virtualization.** Our method works with a large variety of humans and can significantly foster human virtualization procedures. We have conducted tests on individuals wearing clothing, with long hair, or involving clutter from objects, similar to what we can expect from uncontrolled settings and where often the topology significantly deviates from the human body. Several cases are illustrated in Figure 1, and we encourage readers to examine the supplementary materials for extensive qualitative examples. In the supplementary materials, we also demonstrate how our approach can be applied to rough geometries obtained from a user's mobile video using NeRF (Luma AI) to get an avatar ready to be animated. This typically requires extensive human expert annotations and a controlled setting. Our method is robust enough to provide automatic rigging for input obtained from customer devices.

## 6 CONCLUSIONS

In this work, we presented NF-ICP, a refinement method for Neural Fields inspired by ICP principles; a new variant of LVD paradigm that considers human localities; and a registration pipeline robust to disparate challenges. We leveraged a large MoCap dataset for training, and we tested our method in an unprecedented number of datasets, showing robustness and flexibility also on noise and incomplete point clouds, clothing, poses, and all *using the same network weights*. In Supplementary Material, we discuss failure modes and exciting directions for future works. Our method and the release of code and network weights will simplify current 3D registration pipelines, providing a tool useful for several Computer Vision tasks and users, especially for researchers.

## 7 ETHIC STATEMENT

In theory, registration pipelines can be used to acquire identity features and invade individual privacy. However, at present, the input 3D data used in our approach can only be easily obtained with the consent and participation of the target subject. The geometric nature of our self-supervised task helps overcome statistical semantical biases in the training set. NF-ICP could overcome unbalanced representations of subject identities present in the training.

## 8 REPRODUCIBILITY

In section 5, we report information about used data, templates, training time, and involved hardware. In supplementary materials, we list details about architecture, backbones, and training hyperparameters. Our method validation is based on quantitative results on datasets, benchmarks, and leaderboards that are publicly available. Training, validation, and test splits will be publicly available, together with the code and network weights.

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
