# OpenReview forum: "NF-ICP: Neural Field ICP for Robust 3D Human Registration"
_ICLR.cc/2024/Conference — ICLR 2024 Conference Withdrawn Submission_

### Official Review · Reviewer_HArx · 2023-10-25

**Soundness:** 3 good
**Presentation:** 3 good
**Contribution:** 2 fair
**Rating:** 5
**Confidence:** 5

**Summary:**

This paper presents a method to perform 3D human registration from point cloud input. The central idea is to conduct ICP-style iterative refinement between the neural field and the target point cloud after the initial prediction by LVD. Both qualitative and quantitative evaluations on public 3D human registration datasets are presented.

**Strengths:**

- This paper presents a complete working system of 3D human registration from input point cloud to LVD, followed by Neural Field ICP and then final refinement. It also demonstrates interesting applications such as converting the initial scan from Luma AI into an animation-ready avatar driven by the SMPL skeleton.

- The paper presents a wide range of qualitative results from different datasets such as D-FAUST, BEHAVE, and HuMMan.

**Weaknesses:**

- From the perspective of the system, the major technical contribution proposed in this paper, i.e. NF-ICP, is used as a post-processing component in the system. More concretely, the previous pipeline would be LVD (Corona et al. 2022) -> template fitting -> refinement; with NF-ICP added, the pipeline becomes LVD -> NF-ICP -> template fitting -> refinement. While LVD provides the most important initialization, the effect of NF-ICP is relatively incremental. The significance of the contribution in terms of scope seems to be on the minor side.

- From the methodology point of view, the current formulation of NF-ICP evaluates the correspondences and offset only at the target scan points. However, the neural field does not guarantee consistent prediction over different points in the space. Since the vertices are initialized at (0, 0, 0), having an ICP error over the target scan does not guarantee the correctness of offset for points far from the scan surface.

- Since this NF-ICP is a add-on post-processing step, I believe that the central question to discuss is the necessity of NF-ICP in the LVD -> template fitting -> refinement pipeline. Essentially the question here is how much benefit does the additional NF-ICP step brings in comparison with directly using SMPL for the template fitting & ICP, which is the traditional model fitting approach with LVD initialization. On the one hand, the paper gives some quantitative evidence of this by comparing "LVD + R" with "Ours + R" in Table 4 (1.11 -> 1.06, 2.48 -> 2.26) and 6 (6.27 -> 6.13), and Table 3 (3.26 -> 3.08) in the supplementary document, where the improvement seems to be relatively minor.

- On the other hand, some qualitative results are presented beside Table 6, and in Figure 2 & 3 of the supplementary material. However, these examples show that the SMPL model-fitting is under-regularized in the experimental setup of the paper. SMPL encodes a strong prior of human body shape and pose. A well-regularized model fitting process should prevent many of the artifacts from happening. In addition, it is possible to use the LVD initialization together with ICP, instead of completely relying on the LVD correspondences. For such reasons, I believe that the some of the artifacts presented by LVD + R may not necessarily happen under a proper implementation of classical SMPL model fitting.

**Questions:**

- Since my major concern is about the necessity of NF-ICP in a registration pipeline, my central questions for the author is whether more evidence of the advantage of Ours + R over LVD + R can be provided. Here LVD + R should have proper regularization, for example shape regularization by penalizing beta and pose regularization by setting proper prior weight, and balancing between LVD initialization term and ICP term.

- In the FAUST experiment, why is there a setting where input shape is the registration (FaustR)? What is the purpose of this setting?

- Why does this method use 690 vertices instead of 6890 vertices in the original LVD experiment?

- In the LVD paper, their Eq (1) wrote that the neural field g maps a point $\mathbf v_i \in \mathbb R^3$ to $\Delta \mathbf v_i \in \mathbb R^3$, while in their implementation it is actually predicting offset of all the vertices as in this paper (maps $\mathbf v_i \in \mathbb R^3$ to $\Delta \mathbf v \in \mathbb R^{6890 \times 3}$). Although it is more of a fault of the LVD paper, maybe a footnote can be added in this submission to clarify this to avoid confusion for the audience.

---

> ### Author Response · Authors · 2023-11-14
> **Reply to Reviewer HArx - Pt. 1**
>
> We thank the reviewer for the detailed and careful feedback. Here, we try to address the main concerns:
> 1) **the effect of NF-ICP is relatively incremental […] the improvement seems to be relatively minor.**: We respectfully disagree. First, we would consider our merits in the complete set of design choices: 1) Training on a large mo-cap dataset, which provides strong data prior; 2) localized heads, based on LBO spectral clustering on the template, which introduce a geometrical sense; 3) NF-ICP which is the first self-supervised task for neural fields. We proved that all these aspects provide improvements and contribute to reaching an unprecedented quality in shape registration tasks. Secondly, we don't consider the improvement incremental or minor: NF-ICP solidly improves around 10% on top of our LVD version, which already makes use of localized multi-heads and the data provided by the highest high-quality and large dataset available. But even more remarkably, NF-ICP **enables results not possible before**. Supplementary Materials Figures 2, 3, 6, and 8 show extensive out-of-distribution examples where methods without NF-ICP dramatically fail.
> 2) **However, the neural field does not guarantee consistent prediction over different points in the space**: We thank the reviewer for this comment. We encountered this question in our study, and we are happy to validate it with an experiment. We test Ours and Ours+R on FAUST_S dataset, considering three different initializations for LVD: in (0,0,0), on random points of the surface, and random points in the box between -1 and 1 (our shapes are all rescaled in the -0.8 and 0.8). In all the cases, we obtain the same error up to little numerical fluctuation (Ours: 2.55 +- 0.02; Ours+R: 1.86+-0.01). The intuition is simple: while NF-ICP updates the neural field close to the surface, the backbone's data-prior move the points toward the surface, and having a perfect NF far from surface has a limited impact. Once approaching the target, the points meet the updated neural field and converge in the correct location. We will include these observations in the final version of the paper.
> 3) **SMPL model-fitting is under-regularized in the experimental setup of the paper**: We see the reviewer's point, but as stated in Supplementary Section 1.2, we do regularize the SMPL fitting, both in SMPL fitting and during the Chamfer refinement. We do follow the standard implementation of SMPL fitting since our regularizations are identical to the ones of the original LVD paper (L2 regularization weighted 1e-2 for beta parameters, variational prior loss weighted 1e-8 for the pose). Still, we enable the backbone to work in more cases than before, and we cannot see how this can be counted as a weakness. Instead, we would count it as a strength: stronger regularizations could help LVD, but it would limit its expressivity, require careful design, and probably tedious manual tuning. Our approach does not require any of this and works out of the box.
> 4) **it is possible to use the LVD initialization together with ICP**: If our understanding is correct, the reviewer is suggesting adding a regularization to NF-ICP, based on the initial guess of LVD, i.e., to let LVD converge and then penalizing NF-ICP update to deviate from it. We find the proposal interesting, but with a computational drawback: it would require a full convergence of LVD method before applying NF-ICP, to then re-iterate LVD convergence. We agree that many more complex sophistications would be possible and are promising for future exploration that we will mention in our discussion.
> 5) **my central questions for the author is whether more evidence of the advantage of Ours + R over LVD + R can be provided**: We will be happy to provide such evidence. However, we tested on all the datasets we were able to collect and collected extensive quantitative experiments on thousands of real scans. NF-ICP not only always provides a solid improvement in the range of 10% for cases within training distribution (with peaks of around 20%), robustness in the presence of heavy clutter (on BEHAVE, we improve LVD on 80% of the frames, on average by 17.6%, completely unsupervised), but also enables generalization to cases far away from training distribution where LVD does not work at all (partial shapes). We also urge the reviewer to consider our pipeline's other non-trivial choices: localization heads led by spectral clustering and training on the AMASS dataset. We will gladly perform experiments if the reviewer has concrete suggestions to collect the desired evidence.

---

> > ### Author Response · Authors · 2023-11-14
> > **Reply to Reviewer HArx - Pt. 2**
> >
> > 6) **why is there a setting where input shape is the registration (FaustR)?**: We thank the reviewer for raising this point. Faust Registrations are often considered in previous works (e.g., LIE (Marin et al. (2020b)), GeoFMAP (Donati et al. (2020)), Transmatch (Trappolini et al. (2021))) since they provide shapes where the ground truth is directly available on the surface. Such a scenario is ideal for shape-matching approaches (e.g., the Uni baseline). It also give us the occasion to test with shapes at different density (FAUST_R has 6890 vertices per shape, while FAUST_S has around 200K). We will clarify the purpose in the final version of the paper.
> > 7) **Why does this method use 690 vertices instead of 6890 vertices in the original LVD experiment?**: We thank the reviewer for this question. In the original LVD, authors proposed to train on the full SMPL resolution. However, LVD output is a tensor that grows quadratically with the number of template vertices. We decided to train on just a subject of the vertices; this is a pivotal aspect of training on a large-scale dataset like AMASS. On the other hand, our representation lets us recover the full template resolution: 690 vertices are enough to fit a complete template at test time, an operation that needs to be performed in any case in the LVD paradigm. We will include this discussion in the final version of the paper.
> > 8) **a footnote can be added in this submission to clarify this to avoid confusion for the audience.**: We thank the reviewer for this detailed remark; we agree that this could generate confusion and will add the suggested footnote.
> >
> > Please, let us know if our answers satisfy your concerns. We would be happy to provide further clarifications.

---

### Official Review · Reviewer_JdQC · 2023-10-31

**Soundness:** 2 fair
**Presentation:** 3 good
**Contribution:** 3 good
**Rating:** 5
**Confidence:** 3

**Summary:**

This paper is about registration of human meshes. Every vertex of a given a query mesh has to be assigned to a vertex of some target mesh. That is a crucial step in many pipelines for high-quality 3D reconstruction of human bodies. The authors propose a cross-over between a neural field (NF) and the classical iterative closest point (ICP) algorithm. A variant of a previous algorithm (LVD) is proposed. This variant uses multiple output heads that are specialized for different areas of the meshes which requires prior shape segmentation.

**Strengths:**

- registration is a very hard problem that is very important
- the loop between NF and ICP is sound
- a working algorithm for test-time adaption is pretty cool

**Weaknesses:**

My main objections against this paper are in the experimental evaluation:
- There is no evidence how well the algorithm works or whether the performance stems from a better trained NF. The authors should show results of just  s single correspondence estimation (via the NF) without any test-time adaption.
- Similarly, the impact of the multiple output heads is not evaluated. Does the proposed algorithm work with just a single output head?
- The results show shapes after SMPL fitting. This raises the question how much important this last step is? How do results look before this fitting? Are the error metrics computed only on the 690 sample points or on all points of the SMPL shapes?
- What happens if the shape segmentation fails?
- Supplementary, Fig 3: While many of the red points are outlying, the set of points used for optimization might not be same as outlying. Which of the red points have been selected for optimization? How susceptible is the algorithm to outliers?

**Questions:**

See above

---

> ### Author Response · Authors · 2023-11-14
> **Reply to Reviewer JdQC**
>
> We thank the reviewer for the useful feedback. However, we also see several misunderstandings, and we hope we can help to clarify them:
> 1) **Show results of just a single correspondence estimation (via the NF) without any test-time adaption**: we do show several results without test-time adaption. Table 2 shows three different Neural fields without NF-ICP or Chamfer refinement; Table 6 shows a baseline without NF-ICP and Chamfer; Tables 1 and 2 of supplementary show ablation and validation without further refinement.
>
> 2) **The impact of the multiple output heads is not evaluated**: We do evaluate this aspect. Table 2 compares three different baselines, where two (OneShot, LVD_1) have single head; Table 1 of Supplementary is dedicated explicitly to ablate multi-head contribution, proving the usefulness of using 16 heads, especially on real-data; Table 2 of Supplementary carefully considers the difference between different pipeline elements, also considering single vs multiple heads.
> 3) **Results look before SMPL fitting? Are the error metrics computed only on the 690 sample points?**: We thank the reviewer for this question. This aspect has already been discussed in the original LVD paper, and the regularization provided by fitting a template is part of the LVD pipeline itself. It contributes to regularizing the prediction of the backbone, but it is also imperative for evaluation: testing a method using different template resolutions would make the comparison difficult and inconsistent. For this reason, we decided to keep the standard practice of relying on fitting a full SMPL model. Notice that our approach is much more efficient than the approach used by LVD: LVD output matrix grows quadratically with the number of vertices considered for training. Using only 690 vertices is faster, enables training on the large scale of the AMASS dataset, and is enough to fit the SMPL template at inference time – a step that LVD requires in any case.
> 4) **What happens if the shape segmentation fails?**: We are afraid there is a significant misunderstanding here. As reported in Section 4.2, we obtain the regions using spectral segmentation on the template: this is done to design the heads of the network. This is done only once on the template shape before the training. No segmentation is run or required on the test shapes. We will clarify it in the final version of the paper.
> 5) **Supplementary, Fig 3: Which of the red points have been selected for optimization? How susceptible is the algorithm to outliers?**: We thank the reviewer for this question. As reported in Supplementary Section 1.2, we sample at most 20k points, which is performed randomly 20 times. Hence, in the presence of outliers, they are also selected in our process and participate in the unsupervised fine-tuning. Outliers are a concrete challenge for our unsupervised approach, and we consider it in many of our experiments. FAUST Challenge, DYNA, Partial shapes, and especially BEHAVE, all contain significant outliers. Our analysis on BEHAVE provides interesting proof of our robustness to them: despite the point cloud being a fusion of many Kinects full of artifacts, NF-ICP improves the backbone in around 80% of the cases with an average of 17.6% improvement. Our intuition is that the merit is in our NF formulation, which lets us exploit the network's data prior and deal with outliers. We consider our experiments (more than 5k and 10 data sources, with quantitative results on real scans data) quite extensive in considering different nature of data out of distribution, and we also report the limits of our method in the failure cases of Figure 7, Supplementary. If the reviewer has any concrete suggestions for further experiments, we'll be happy to consider them for the final version of the paper.
>
> Please let us know if our answers satisfy your concerns. We would be happy to provide further clarifications.

---

### Official Review · Reviewer_FtZ7 · 2023-11-01

**Soundness:** 3 good
**Presentation:** 2 fair
**Contribution:** 2 fair
**Rating:** 5
**Confidence:** 4

**Summary:**

This paper proposes a neural field (NF) refinement method using ICP principles and a pipeline for human body registration using a new variant of learned vertex descent (LVD). Experimental results validate the design and robust performance.

**Strengths:**

- The overall design is intuitive and straightforward, and the method is easy to replicate.

- Ablation experiments quantitatively demonstrate the design of this method.

- The experiments show good generalization to real challenging data.

**Weaknesses:**

- By reading the method part of Section 4, including the main content of the method in Section 4.1 and the implementation in Section 4.2, I feel that there are limited new and engaging contributions.
This method is like a new application of existing methods, Neural Fields (NF) and Learning Vertex Descent (LVD).
This reliance on existing works may raise concerns about the substantial contribution of the paper.

- In Table 3, the performance of this method after refinement is greatly improved. What would be the result if the refinement was not performed in Tables 4 and 5? What would be the performance if other baseline methods also used the refinement step?
This makes me concerned about the real performance of the method.

- Why is there no comparison with more methods in the BEHAVE data set in Table 6?

- How efficient is this method compared to baseline methods?

- The symbols NFICP and NF-ICP are mixed, which is not standardized.

**Questions:**

Please address the concerns in the Weaknesses part.

---

> ### Author Response · Authors · 2023-11-14
> **Reply to Reviewer FtZ7**
>
> We thank the reviewer for the valuable feedback, and we are happy to reply to the raised comments:
> 1) **New application of existing methods**: we respectfully disagree; we recognize the use of some existing techniques, but we would highlight that we have significant design changes. We are the first method to train on the full AMASS dataset; we propose localized LVD variants based on spectral segmentation, which show a concrete gain; we propose a self-supervised task for neural fields, which is the first of its kind.
> 2) **Results without Refinement**: We agree that Chamfer refinement contributes to the final quality of the method since it helps to bridge the gap between the data prior and the geometry. However, to converge at the right solution, it requires to be close to the correct minimum. We show on BEHAVE (Table 6) and DYNA (Table 3, Supplementary) and on a public benchmark (Table 4) the difference with and without NF-ICP, showing how our method has a concrete contribution to the final results (in a measure that is stably around 10%, but it has even peaks around 20%). Chamfer refinement is a standard in shape registration community, and it is what enables a fair comparison in Tables 4 and 5, where many other methods use test-time Chamfer refinement ((Bhatnagar et al. (2020)), (Deprelle et al. (2019)), (Groueix et al. (2018)), (Trappolini et al. (2021))) or other post-processing refinements (ZoomOut in Donati et al. (2020))). If the reviewer finds it informative, we will process pre-chamfer results and add them to the final version of the paper.
>
> 3) **Comparison on BEHAVE**: We thank the reviewer for the suggestion and understand how it would strengthen our validation setting. Our paper considers different approaches of different natures trained on the same dataset (OneShot, LVD_1, LVD_16, Uni-60, Uni-2070), and in Table 2 and Table 3 we demonstrate the superiority of LVD_16. BEHAVE results (Table 6) show how NF-ICP improve the results even in cases of extreme noise condition, and we were interested in showing it on the best backbone available. Also, Uni-60 and Uni-2070 are not directly applicable since they look for correspondence on the point cloud and couldn’t generate the underlying template. Testing the baselines on BEHAVE would require many days since we consider thousands of dense point clouds, and we cannot provide the results within the time of this rebuttal. We appreciate the suggestion, and for the final version of the paper, we will also include the results for OneShot and LVD_1.
> 4) **Efficiency compared to baseline methods**: Thanks for the question; as reported in the “Computational Timing” paragraph, our full registration pipeline requires around 60 seconds. Baselines based on LVD without NF-ICP would need a couple of seconds less. Uni baselines are required to compute the nearest neighbor in a high-dimensional space for all the input points, which can be hundreds of thousands in the case of real scans. This must be done independently for each pair of shapes we want to match. As a reference, for Ours+R, completing the evaluation on FAUST_S dataset requires around 2 hours (100 shapes registration + 100 pairs matching), while for Uni-2070 it takes about 4 hours.
> 5) **The symbols NFICP and NF-ICP are mixed**: Thanks for this remark; we’ll fix this in the final version of the paper.
>
> Please let us know if our answers satisfy your concerns. We would be happy to provide further clarifications.

---

### Author Response · Authors · 2023-11-14
**General Reply**

We thank all the reviewers for their insightful feedback and the time dedicated to reviewing our work.

We are glad that they found our design **intuitive, straightforward, easy to replicate** (R1), **sound and cool** (R2), which validity and generality are proved by **extensive experiments** (R1, R3), also demonstrated by **interesting applications** (R3), and providing overall a **complete working system of 3D human registration, which is a difficult task** (R1, R3).

Despite reading **a consensus in recognizing the soundness and exciting results** provided by our method, we are displeased with such low scores, and we hope our rebuttal can help the reviewer reconsider their evaluation.

We reply to every reviewer individually, but we want to further emphasize what we consider our work's main achievements.

We obtain a robust human registration pipeline, getting a generalization on many challenges as no other method has been capable before. We outperform competitors on public benchmarks, and we achieve this by non-trivial technical steps:
1) **training for the first time on AMASS**, a high-quality large motion-captured dataset;
2) providing a **localized modification for LVD** based on LBO spectral theory;
3) introducing **the first unsupervised self-supervised finetuning** for neural fields.

Our technical contribution has been validated with a thoughtful ablation study (please revise our supplementary materials), tested on different domains, and we are unaware of previous works that provided a similar set of empirical evidence (i.e., more than ten datasets different from the training distribution).

We finally urge the reviewers to consider **the impact of releasing our registration pipeline**, which will serve as a useful tool and have significant immediate applications.